# Percutaneous Fluoroscopic-Guided Celiac Plexus Approach: Results in a Pig Cadaveric Model

**DOI:** 10.3390/ani14101478

**Published:** 2024-05-16

**Authors:** Francesco Aprea, Yolanda Millan, Anna Tomás, Gemma Sempere Campello, Rocio Navarrete Calvo, Maria del Mar Granados

**Affiliations:** 1Hospital Veterinari Canis Mallorca, 07013 Palma, Spain; 2Department of Comparative Pathology, School of Veterinary Medicine, University of Cordoba, 14071 Cordoba, Spain; 3Instituto de Investigación Sanitaria de las Islas Baleares (IdISBa), 07120 Palma, Spain; anna.tomas@ssib.es (A.T.); gemma.sempere@ssib.es (G.S.C.); 4Animal Medicine and Surgery Department, School of Veterinary Medicine, University of Cordoba, 14071 Cordoba, Spain; nacar6@hotmail.com (R.N.C.); pv2grmam@uco.es (M.d.M.G.)

**Keywords:** regional anaesthesia, celiac plexus, visceral pain, pigs, fluoroscopy

## Abstract

**Simple Summary:**

The celiac plexus (CP) is a dense network of ganglia and sensitive fibres receiving nociceptive inputs from most abdominal viscera. In human medicine, the CP is a therapeutic target to control pain originating from pancreatic tumours and chronic pancreatitis. The fluoroscopic-guided block and neurolysis of the CP are commonly used in human analgesia but they have not been described in animals. We describe a fluoroscopically guided transcutaneous approach to the CP in swine cadavers to assess the feasibility of the technique in veterinary species. This technique is shown to be feasible, and its application in veterinary subjects suffering abdominal pain should be assessed.

**Abstract:**

Celiac plexus block (CPB) and neurolysis (CPN) are used for pain management in people suffering from abdominal tumours or chronic pancreatitis. The fluoroscopically guided approach common in human medicine has not been described in veterinary settings. The aim of this study was to describe a fluoroscopic approach to the celiac plexus (CP) in fresh pig cadavers. Twelve animals were included in the procedure. Cadavers were positioned in sternal position and, under fluoroscopic guidance, a Chiba needle was inserted parasagittal at 6 cm from the spinal midline at the level of the last thoracic vertebra. From the left side, the needle was directed medio-ventrally with a 45° angle towards the T15 vertebral body; once the vertebral body was contacted, the needle was advanced 1 cm ventrally towards the midline. Iodinated contrast was injected to confirm the location. Following this, 2 mL of dye (China ink) was injected. A laparotomy was performed, and dyed tissue was dissected and prepared for both histochemical and immunohistochemical techniques. In 10 out of 12 samples submitted for histological evaluation, nervous tissue belonging to CP was observed. Fluoroscopy guidance allows for feasible access to the CP in swine cadavers in this study. Further studies are warranted to determine the efficacy of this technique in swine and other veterinary species.

## 1. Introduction

Control of visceral pain is challenging in both human and veterinary medicine [1,2,3,4]. Differently from somatic pain, visceral pain is diffused, poorly localized, and it is associated with a vegetative autonomic response. Visceral nociception causes pain referred to other locations due to viscero-somatic and viscero-visceral synaptic convergences [4,5]. The autonomic nervous system contributes to the sensory innervation of viscera; sympathetic fibres run along sensitive afferences (mainly C fibres) and play a major role in the transmission of visceral pain [4,5].

In small animals, painful conditions such as pancreatitis and pancreatic neoplasia are common and associated with severe visceral pain [2,3]. In people, systemic analgesics do not always control severe pain; thus, interventional techniques have been employed to improve pain management [1].

The celiac plexus (CP) is a dense network of ganglia (paired celiac ganglia, unpaired cranial mesenteric ganglia, and paired aorticorenal ganglia), interconnecting preganglionic sympathetic and parasympathetic efferent, and visceral sensory afferent fibres. The CP is the largest visceral plexus, receiving nociceptive inputs from the liver, gall bladder, pancreas, spleen, adrenal glands, kidney, distal oesophagus, and intestine [4,5,6].

In humans, the CP lays in the epigastrium within the retroperitoneal fat on the antero-lateral surface of the aorta, in proximity to the celiac trunk just before the emergence of the renal arteries. It stands posterior to the stomach, pancreas, and kidneys and inferior to the liver and diaphragmatic pillars. The diaphragmatic crura separates the CP from the spinal cord. In humans, the mean size of the CP is 2.7 cm, and it is in proximity to the 12th thoracic vertebra or at thoracolumbar level, but individual variations are possible [6,7]; in dogs and horses, the CP is found ventrally to the first lumbar vertebra [8,9].

The CP block (CPB) and neurolysis (CPN), first described in 1914 by Kappis [10], have been used for decades in human analgesia to treat pain associated with chronic pancreatitis and abdominal malignancies such as pancreatic cancer [1].

In veterinary medicine, such options are lacking. Implementing the interventional analgesic technique in veterinary pain management could improve analgesia and enhance comfort in subjects suffering from refractory visceral pain. In humans, fluoroscopy, ultrasonography (US), and computed tomography (CT) have been used to perform CPB and CPN, but none of these techniques have been considered superior to another in terms of efficacy [7,11].

In a porcine model, laparoscopic [12] and endoscopic ultrasound-guided [13,14] approaches to CP have been described in experimental settings, but a transcutaneous fluoroscopic approach, commonly used in clinical settings in humans [15], has not been described in veterinary medicine yet.

The aim of this study was to describe a transcutaneous fluoroscopic-guided approach to the CP in fresh pig cadavers.

## 2. Materials and Methods

This cadaveric study obtained institutional ethical approval (CEEA/002/2019).

### 2.1. Animals

Pigs were enrolled in the study from July 2019 to December 2020. One pig cadaver was used for preliminary anatomical study. Animals were obtained from an unrelated, no-survival study, and no animal was euthanized. Pigs were euthanised humanely by intravenous injection of 5 mL of pentobarbital 20% (Dolethal, Vetoquinol, Madrid, Spain) for this project. All pigs belonged to the same breed (Duroc cross Large White and Landrace). If the procedure could not be carried out immediately after euthanasia, animals were excluded from the study.

### 2.2. Preliminary Anatomical Description and Fluoroscopic Technique Development

Due to breed-related anatomical differences in swine, with special reference to the number of thoracic vertebras, a cadaver, not included in the procedure, was used as a preliminary anatomical model to determine the number of vertebras, fluoroscopic landmarks, and the location of the CP. After identifying the correct location of the CP in the pilot subject (confirmed by histological evaluation), we carried out the technique on the other cadavers included in the study.

Cadavers were positioned under the fluoroscopy C-arm (BV 300, Philips, Amsterdam, The Netherland) in sternal recumbency on a radiolucent table (Figure 1).

A posterior para-aortic approach to the CP, as described in humans, was employed [7]. Under fluoroscopy, using a dorso-ventral radiological beam (40 kV, 0.10 mA), the midline of the spine, the vertebral bodies, and the last rib were identified and marked (Figure 2).

A 22 G, 15 cm Chiba fine needle (BD, Franklin Lakes, NJ, USA) was inserted parasagittal at the left side of the spine, at 6 cm from the midline at the level of the last thoracic vertebra. The needle was advanced medio-ventrally with a 45° angle towards the ventral surface of T15 vertebral body. Once the vertebral body was contacted, the needle was advanced 1 cm further ventrally towards the midline into the paravertebral fascial plane (Figure 3). Additionally, a latero-lateral fluoroscopic view of the needle was produced. Once the position was considered appropriate, 5 to 10 mL of iodinated contrast (Visipaque 320 mg/mL; GE Healthcare, Chicago, IL, USA) was injected and its pattern of distribution assessed. If intravascular injection was suspected, the needle was gently withdrawn, and the administration of contrast media was repeated until spread of the contrast media was considered appropriate (retroperitoneal diffusion). Once the needle was in a correct position, 2 mL of dye (China Ink, Pelikan Drawing Ink., Hanover, Germany) was injected.

### 2.3. Tissue Collection, Processing, and Histological and Immunocytochemistry Studies

Through laparotomy, the abdominal cavity was explored, and an anatomical dissection of the dyed area was performed to identify the stained tissue (Figure 4).

All samples (2 cm, Figure 5) were placed in histological cassettes and fixed in 4% neutral-buffered formalin over 24–72 h and embedded in paraffin wax for histological study. A total of 3 blocks were obtained for each pig. Histochemical study was performed in tissue sections of 4–5 µm from each paraffin block and stained with haematoxylin–eosin (HE). Furthermore, tissue sections of 4 µm thick were placed on Vectabond-coated slides (Sigma Diagnostics, St. Louis, MO, USA) for immunohistochemical study. Specific neuronal markers as neurofilament (NF) and glial fibrillar acid-protein (GFAP) were used for neuronal tissue enhancement.

For immunohistochemical study, the slides were deparaffined, rehydrated in a graded series of alcohol, and incubated with 3% hydrogen peroxide in methanol for 30 min. Heat-induced antigen retrieval was performed in a 20 min water bath at 98 °C with 0.01 M citrate buffer (pH 6.0) for neurofilament (NF) antibody, exclusively; after cooling for about 30 min at room temperature, sections were covered with 10% normal goat serum in phosphate-buffered saline for 30 min before incubation with one of the primary antibodies for 18 h at 4 °C (1:50 for GFAP and 1:75 for NF). Afterwards, the avidin–biotin-peroxidase complex method was applied, as recommended by the manufacturer (Vector Laboratories, Burlingame, CA, USA). Three,3-diaminobenzidine tetrahydrochloride (DAB) (Sigma Diagnostics, St. Louis, MO, USA) for NF and GFAP were used as chromogens. The slides were counterstained with Harris haematoxylin. As positive control tissues, porcine spinal cord tissue was used for NF and GFAP antibodies.

In the tissue sections stained with HE, presence of CP was established when clusters of neuron cell bodies (loosely encapsulated by ganglionic gliocytes) and/or a network of nerve fibres of different calibres (in at least from 1 to 3 block per animals) were identified histologically.

## 3. Results

### 3.1. Preliminary Anatomical Description and Fluoroscopic Technique Development

The cadaver used as the anatomical model presented 15 thoracic vertebras. Following celiotomy, the CP was localized retroperitoneally at the level of the last thoracic vertebra (T15) in proximity to the celiac and mesenteric arteries originating from the abdominal aorta, caudally to the diaphragmatic pillars and cranial and dorsal to the kidneys.

A total of 12 pig cadavers were enrolled for the procedure, and all of them presented 15 thoracic vertebrae. The body mass was between 25 and 30 kg, and subjects were 3 months old and female. The described transcutaneous dorsal para-aortic approach to the CP was successfully carried out in all subjects.

### 3.2. Gross Anatomical and Microscopic Studies

Following laparotomy, in all cadavers, the dye was visible in the retroperitoneal space in proximity to the celiac trunk. Stained tissues included other retroperitoneal structures: cranial mesenteric and celiac arteries, aorta, adrenal gland, muscle, and fat. Spreading ranged from 2 to 4 cm in length. The nervous structures (fibres and/or ganglia) were not recognizable macroscopically. Dissection and collection of a sample of 2 cm of the dyed tissue were performed in all 12 cadavers.

Microscopically with HE, neuronal tissues belonging to the CP were observed in 10 out of 12 samples (one for animal). Specifically, in six animals, clusters of cell bodies of neurons and nerve fibres of different sizes were identified. In one animal only, a small ganglion was identified, and in the other three animals, a network of myelinated nerve fibres was observed (Figure 5). In the samples obtained from two cadavers, neither ganglia nor myelinated nerve fibres were observed with HE or IHC studies.

Other structures were histologically identified in the examined samples, including blood vessels, adrenal gland, adipose, muscular tissues, lymph node, and osseous spicules.

The IHC study, using NF and GAFP, confirmed the histochemical results; brown cytoplasmic staining was observed in cell bodies of neurons and nerve fibres with NF and GAFP, respectively (Figure 6).

## 4. Discussion

This study is the first that describes a percutaneous fluoroscopic approach to the CP in a cadaveric porcine model. This approach was shown to be feasible and could be translated, with appropriate modifications according to species, in both experimental and clinical settings.

In humans, pain is reported in 95% of subjects suffering with biliary-pancreatic tumours, and commonly administered analgesic drugs are not effective to treat nociception associated with this condition [11]. Opioids have not been associated with increased complications in acute settings of pancreatitis [16], but the use of systemic morphine is controversial because, in humans, the pure mu agonists cause Oddi sphincter contractions, and an increase in pressure in the bile duct might worsen the visceral pain [17]. This is not demonstrated in veterinary species but should be considered. Also, in people, undesirable effects can be seen with prolonged opioid administration [18]. In dogs undergoing orthopaedic surgery, those receiving post-operative methadone every 4 h had increased chances of vomit, vocalization, and reduced food intake when compared to dogs receiving methadone only if an increase in pain score was detected [19].

In canines, pancreatitis is described as the most painful pathology affecting the abdomen, leading to excruciating pain [3,20]. Differently from post-surgical or chronic (osteoarthritic) pain, research on non-surgical visceral pain and validated tools to assess it are lacking in veterinary medicine [2]. In a descriptive survey among veterinary surgeons, pancreatitis was identified as the most frequent cause of abdominal pain in dogs, and pancreatic tumours cause severe visceral pain in small animals [3]. The suggested analgesic treatment in canines includes opioids (systemic or neuraxial administration), ketamine and lidocaine intravenous infusions, and low doses of corticosteroids [20]. Even though these drugs are employed on a daily base in small animal practice, the level of evidence to support their use for the treatment of severe visceral pain is low, and clinical trials are lacking [20]. As for somatic pain, persistent visceral pain leads to chronic abdominal pain and central (supraspinal) neuroplastic changes that further contribute to facilitating and perpetuating the visceral nociceptive input, leading to peripheral and central sensitization [21].

In people, the use of regional anaesthesia techniques is considered crucial to provide appropriate pain management [22]. This is in accordance with a modified World Health Organization analgesic ladder, which includes interventional procedures as a part of a multimodal analgesic treatment [22].

The fluoroscopically assisted CPB and CPN are common procedures in human analgesia to treat refractory visceral pain [7,11], but they have not been reported in any veterinary species.

Different fluoroscopically guided techniques have been described to approach the CP in humans, including posterior para-aortic, anterior para-aortic, trans intervertebral disc, and trans-aortic [7,23,24,25]. Neither seems to be superior to another in terms of efficacy, but the posterior (dorsal) technique seems safer [7,23,24,25]. Injecting bilaterally using similar approaches did not produce superior analgesia than a one-side injection [26,27]. The dorsal para-aortic approach is the most used [15], and it is the fluoroscopically guided technique routinely employed by the human interventional radiologist who took part in this study.

The porcine model has been an accepted translational model for humans for decades [28], but even though the transcutaneous approach is the most used in humans [15], only laparoscopic [12] and endoscopic ultrasound-guided [13,14] techniques to block the CP have been described in experimental settings in this species.

The laparoscopic approach is indicated during staging laparoscopy in the case of unresectable neoplasia [12]. The endoscopic ultrasound approach is promising, but it is mainly employed by gastroenterologist rather than interventional pain management practitioners [13,14]. In both techniques, the major arteries are used as landmarks (celiac trunk and para-aortic region) for CP rather than bony structures. In these studies, no anatomical description of the CP location in swine was available [12,13,14].

We decided to use a porcine model for the following reasons:-Availability of fresh cadavers; with respect to the 3Rs principle, subjects used in this study were obtained from an unrelated no-survival study, and no animal was euthanized for this purpose.-The similarity of the swine spine anatomy to that of humans facilitated the execution of the technique, as described in people.-In the last few decades, many regional analgesia techniques described in human medicine have been translated into veterinary medicine following cadaveric descriptive studies [28]. Following refinement, the cadaveric techniques have been largely employed in clinical settings for pain management in different target species.-The supervision of an interventional radiologist (GS) familiar with the execution of this technique in people would not have been possible outside the research facility using another species.

Ravasio et al. 2014 [8], in a cadaveric study in dogs, successfully stained the celiac plexus using a blind technique in four out of five animals. Differently from our protocol, cadavers were positioned in the right lateral recumbency, and a spinal needle was inserted between the first and second lumbar vertebra parallel to the transverse process. The volume used ranged from 3 to 15 mL (China ink) depending on dogs’ body mass (1.9–31 kg). In this study, a loss of resistance to injection was used to identify the retroperitoneal location prior to dye administration. Similarly to our findings, in this report, in addition to CP, histological examination of the stained tissue included other structures, such as the vessels (celiac and mesenteric arteries and cava), adrenal glands, and muscles. To reduce the volume of injectate, Ravasio et al. [8] concluded that fluoroscopic- or ultrasound-guided techniques should be investigated.

The same group evaluated the clinical efficacy of an ultrasound-guided CP block in equine suffering from ileus [9]. As in dogs, the needle was inserted between the transverse process of L1–L2 parallel to the spinus process and advanced dorsoventrally; loss of resistance was used to confirm the retroperitoneal location. Following the administration of 50 mL of a mixture of lidocaine and ropivacaine, an improvement in gut motility was visible in seven out of nine subjects. Prescence of sweating on the animal flank confirmed sympathetic blockade [9].

In both studies, spinal needles were used for the injection [8,9]. In humans, Chiba fine needles are commonly used for interventional analgesia techniques and specifically for CPN [29]. This allows one to reach the target structure without any problems.

The technique we described did not present special challenges. Ribs and vertebral bodies were used as landmarks in all cadavers; accessibility and visibility of the landmarks such as spinal cord and ribs were considered good. In all cases, the needle had to be slightly redirected to achieve a good imaging of the contrast medium spreading within the retroperitoneal space, excluding intramuscular or intravenous administration.

In 10 out of 12 cadavers, neuronal tissue belonging to CP (axons and/or ganglia) was stained. The entire CP was not visualized or dissected, but fibres, ganglia, or both were identified in 10 subjects.

Due to the cadaveric nature of this study, the clinical utility of the block was not tested, but the described method permitted a feasible approach to the target structure. Fluoroscopy permits clear identification of the bony structures used as anatomical landmarks (ribs, vertebral bodies), facilitating the learning curve of untrained operators. Soft tissue structures such as vessels or organs are not recognized under fluoroscopy, and this is why radio opaque contrast is injected to further guide needle placement. In living subjects, a negative aspiration test should always be performed prior to injection, as in any local analgesic technique to avoid accidental intravascular administration. Even though accidental vessel puncture is a possibility, this should not lead to major complications in subjects with normal coagulative function. In humans, a transaortic technique for CP neurolysis, which implies transfixing the aorta to reach the CP, has been described for routine clinical use [7,25].

As the CP is located at the thoracolumbar level in most veterinary species, the described imaging-guided technique could be translated to other species for clinical purposes. In this case, we did not assess the ideal volume of injectate to be used for a clinical block, but larger volumes than 2 mL such as those described by Ravasio et al. [8,9] could be employed for pilot descriptive studies in small animals.

In humans, complications associated with CPB and CPN include visceral and vascular injury due to the transfixion of vessels or organs, and the most common undesired effects reported are weakness, lower chest pain, postural hypotension, failure of ejaculation, difficult urination, and heavy leg sensation [7,15]. Empyema, retroperitoneal abscess, pneumothorax, haematuria, peritonitis, and pulmonary embolism have also been reported as a consequence of needle transfixion but are considered rare [7,15].

Using a cadaveric model, we could not assess the incidence of unwanted events or side effects; on macroscopic examination carried out during exploratory celiotomy, we did not find major vessel or organ damage due to accidental transfixion of the needle, but this was not the aim of this study.

The retroperitoneal space creates a delimited anatomical area (rich in retroperitoneal fat) that eases the spread of the injectate, avoiding its distribution within the abdominal cavity, but, still, accidental punctures of other structures can occur. On dissection, and tissue sampling, we excluded intravascular injection of the dye looking at the inner endothelial wall. When CPN with alcohol is performed, care not to spread the solution to other anatomical locations is crucial to reduce unwanted effects. When local anaesthetics are used, caution to avoid intravascular injection is mandatory, but no other nervous structure than the CP is likely to be blocked.

When executing CPB and CPN in veterinary subjects, the clinician should discuss the possible complications related to the technique. It is important to take into consideration that subjects undergoing these interventional procedures are suffering severe pain and do not respond to conventional pain management; therefore, the benefits (related to superior comfort) often outweigh the possible (rare) unwanted effects.

Other described techniques used in humans for CPB such as US- and CT-guided ones seem to reduce incidence of accidental vascular and organ puncture compared to fluoroscopy, but there is currently no supporting evidence, and none of the techniques have been described superior to another in terms of efficacy [7,11]. An advantage when using US-guided approaches is the absence of exposure to radiation for both the patient and operator.

Cadaveric studies aim to describe a technique considering interspecific anatomical differences for future clinical applications in different species. Pigs have more thoracic vertebras than other mammals, such humans and dogs; this could lead to anatomical variations in the CP localization. A pilot anatomical study was performed on a cadaver of the same breed and genetic line to identify the anatomical location of the CP. As described in humans [7], variation in the position of the CP can be observed within the same species. In the two subjects where no neuronal tissue (ganglia or fibres) was sampled, the CP might have been localized more caudally at the lumbar level. Inserting the needle at the L1–L2 level (as described in humans, dogs, and horses) or using a larger volume of dye could result in CP staining. In the two cadavers where neuronal tissue was not found, poor dissection, larger presence of perivascular fat, and connective tissue in the collected sample might have played a role too.

Being a descriptive cadaveric study, we did not look at the efficacy of the technique. In humans, the clinical effectiveness of CPB or CPN depends on operator, timing, and outcome measures. In a meta-analysis, CPN was reported to provide good to excellent pain relief in 89% of the subjects during the first two weeks following the procedure, partial to complete analgesia continued in 90% of the subjects following 3 months, and in 70 to 90% of the subjects, benefits were observed until death [30,31]. In small animals suffering pancreatic tumours or chronic pancreatitis, similar results might be observed once the technique is refined. When a single central injection was compared to bilateral administration during a posterior para-aortic approach, no differences were reported for the duration of analgesia or reduction in systemic analgesic consumption [11]; this is why we decided to employ a unilateral technique.

Recently, a less-invasive procedure, the ultrasound-guided quadratus lumborum block (QLB), has been described in a cadaveric model in dogs [32]; this technique seems to provide visceral analgesia, blocking the sympathetic fibres contributing to visceral pain transmission [33]. The QLB might be useful to provide visceral analgesia in perioperative settings, but differently from the CP, it has not been described for chronic or medical pain management in any species. The CP is, anatomically, the main network for the nociceptive transmission of sympathetic afferences coming from the cranial abdominal organs, thus remaining the main therapeutic target [7].

This study presents several limitations:

We assessed anatomy (and technique) in a specific swine breed, so other breeds might have anatomical differences, and the technique might need modification. In most species, the CP is located at the thoraco-lumbar level, and this region can be easily recognized under fluoroscopy due to the presence of ribs, but a modified approach (more caudal) to the CP might be more appropriate in other swine or other species.

The CP was not easily recognized macroscopically, as it was surrounded by connective and adipose tissue present in the retroperitoneal space surrounding the aorta and its branches. The submitted sample included the vessel wall and surrounding tissue. At histological examination, other tissues not belonging to the CP were observed, including glandular (adrenal), lymphatic, adipose, bone, and muscular tissue. This might be related to the proximity of these structures to the plexus, gross dissection, and to the impossibility of macroscopically distinguished types of stained tissues during sampling. The presence of osseus spicules and muscle in some preparations might be due to iatrogenic contamination of the sample when advancing the needle or injecting after contacting the vertebral body and transfixing the muscle layer.

A thorough dissection of the CP was not performed, but it would have been desirable. It was not assessed if the plexus was entirely or only partially stained. We only confirmed the presence of nervous fibres and/or ganglia belonging to the CP in the submitted samples. The lack of complete diffusion through the whole CP might result in a “patchy” block and incomplete visceral analgesia. In this study, the most appropriate volume of injectate to be used for the technique was not established. It would be interesting to compare different aliquots to determine the most effective volume to block the CP under fluoroscopy guidance. We decided to use only 2 mL of dye to have a localized stained tissue, facilitating collection and histological evaluation. In clinical settings, larger volumes are used for CPB (increasing the chances of the block), even though the use of smaller volumes is advocated for neurolysis [33]. As previously described, the spreading of the dye differs from the one of local anaesthetic or other (neurolytic) solutions [34]. Specifically, long-lasting local anaesthetics followed by steroids (e.g., triamcinolone) or dehydrated ethyl alcohol are usually employed in humans for the CPB and CPN, respectively [7,35]. The specific spreading of these substances (strictly related with clinical effects) was not assessed in our model.

We refer especially to human literature because the CPB and CPN have not been described for clinical use in veterinary species, and this is another limitation. Studies published on pigs used this animal as a translational model only, and no details about their anatomy or clinical findings were provided.

Further research is warranted to establish the role of CPB and CPN in veterinary clinical practice.

## 5. Conclusions

Our results show that the percutaneous fluoroscopic approach to CP is a feasible technique in a porcine model. Considering that visceral pain management is challenging, CPB and CPN via the percutaneous fluoroscopic approach might be used to provide visceral analgesia in swine and other species. Clinical studies are warranted to determine the effectiveness of fluoroscopic-guided CPB and CPN in veterinary visceral pain management.

## Figures and Tables

**Figure 1 animals-14-01478-f001:**
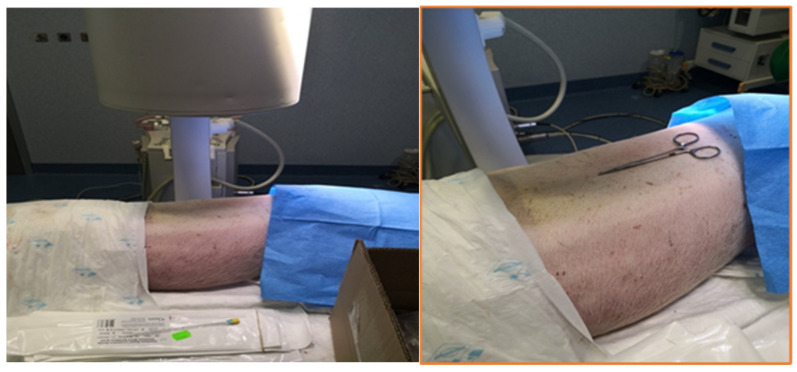
Positioning of the pig cadaver in sternal recumbency under fluoroscopy C-arm on a radio transparent table.

**Figure 2 animals-14-01478-f002:**
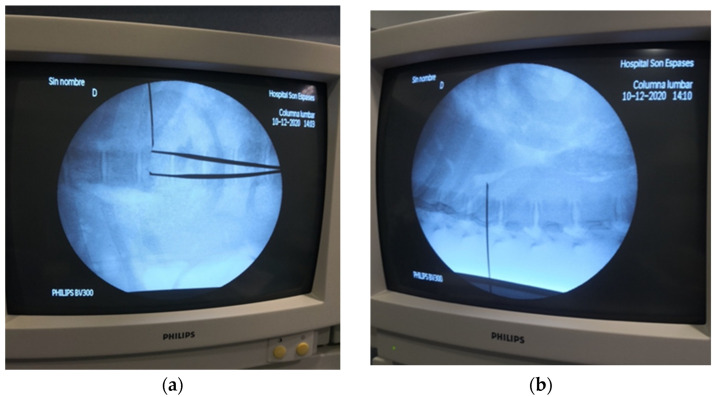
Identification of last thoracic vertebra as a landmark for transcutaneous dorsal para-aortic approach to the celiac plexus (CP) in pig cadavers, dorso-ventral (**a**) and lateral (**b**) views.

**Figure 3 animals-14-01478-f003:**
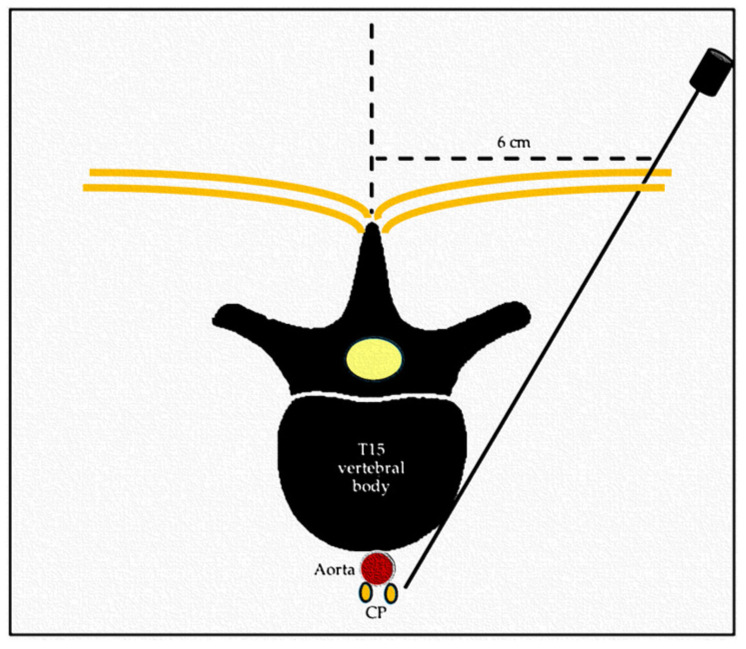
Schematic representation of the dorsal transcutaneous para-aortic approach to the celiac plexus (CP). Transversal view of the spine at thoracolumbar level.

**Figure 4 animals-14-01478-f004:**
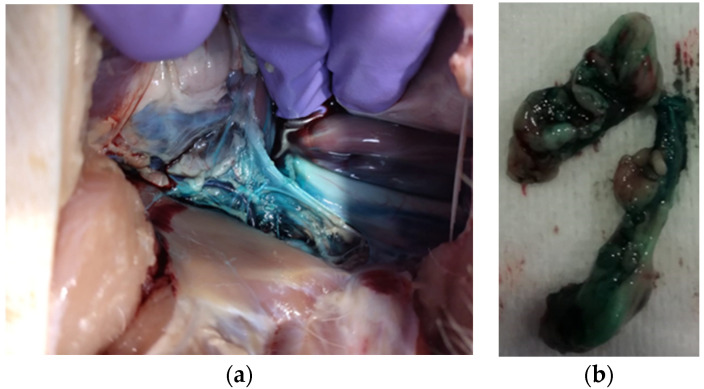
Retroperitoneal stained tissue visible following exploratory laparotomy (**a**) and 2 cm samples dissected for histological preparation (**b**).

**Figure 5 animals-14-01478-f005:**
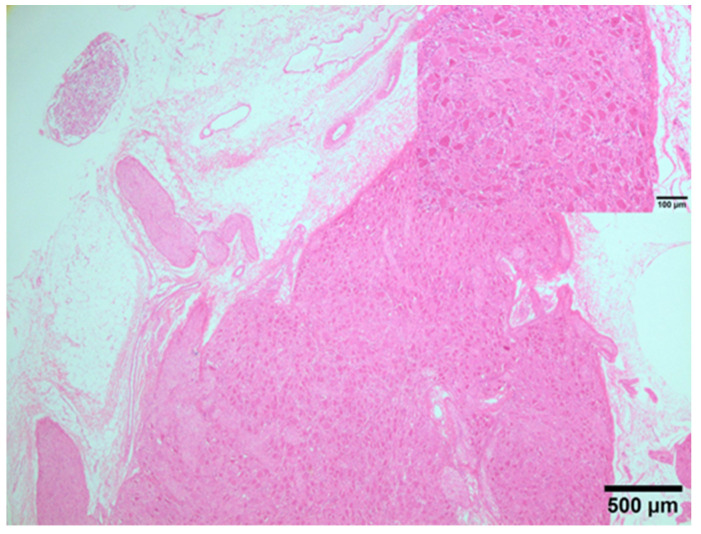
Ganglion from swine CP; clusters of neuron cell bodies loosely encapsulated by ganglionic gliocytes and some nerve fibres of different calibres. Insert: details of neuron cell bodies. HE. 2×.

**Figure 6 animals-14-01478-f006:**
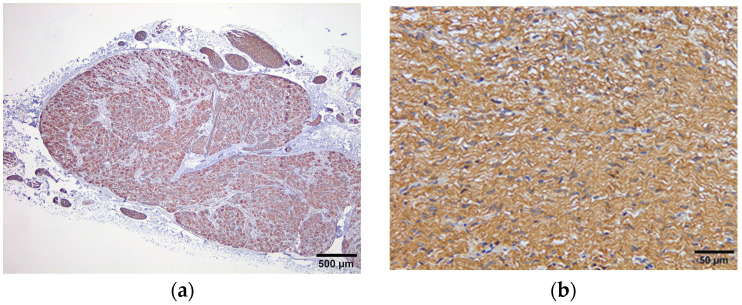
Ganglion from swine CP showing immunohistochemical expression of NF in the cytoplasm of neuron cell bodies (**a**); nerve fibres from swine CP showing immune expression of GFAP (**b**).

## Data Availability

Further data are available on request from the corresponding authors.

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
