# Peer review of "Percutaneous Fluoroscopic-Guided Celiac Plexus Approach: Results in a Pig Cadaveric Model"

_animals, 2024, doi:10.3390/ani14101478_

Round 1
Reviewer 1 Report
Comments and Suggestions for Authors
This is an interesting , well conducted study with potential application of the technique in relatively smaller species of domestic animals.
The paper is far too long and requires extensive editing
Comments on the Quality of English LanguageThere are a number of Americanisation spellings whic heed attention
Author Response
Dear reviewer,
Thank you very much for your comments.
We will proceed with the suggested modifications.
Best regards
The authors
Comments and Suggestions for Authors
This is an interesting , well conducted study with potential application of the technique in relatively smaller species of domestic animals.
- Thank you very much, we are glad you think the study is interesting and with possible clinical application.
The paper is far too long and requires extensive editing.
- Thanks to point it out. We will discuss this issue with the editor to better understand which parts can be removed or edited.
There are a number of Americanisation spellings which seek attention.
- Thanks to point this out. The text has been modified into British English.
Thank you so much for help
Reviewer 2 Report
Comments and Suggestions for Authors
Dear authors, thank you for the paper.
I read the paper with great pleasure. I have a few notes that I hope will provide insights to implement the quality of the work.
INTRODUCTION
LINE 40: typing error (diffuse instead of diffused)
LINE 60: phrases could be linked better
MATERIALS AND METHODS
2.1 PARAGRAPH: here are presented some results, for example the age and weight of the animals enrolled. I think that instead it should report some exclusion criteria.
LINE 102: if the fluoroscopy used a dorso-ventral radiological beam why the position of the needle is confirmed by a latero-lateral fluoroscopic view?
LINE 103: what caused the variations in the amount of contrast injected?
LINE 113: is not clear how the samples were chosen
LINE 133: typing error (staining instead of stained)
LINE 133-136: I would rephrase this section, as it is not very clear.
RESULTS
LINE 152: which direction followed the stain
I would not put all the images as a subchapter of the results. Some of them are more related to the method part.
DISCUSSION
LINE 196: there is some literature on the effects of prolonged opioids treatment also in veterinary medicine, maybe cite some of it.
LINE 198-200: I don’t see how this information are useful in this context
LINE 261: Following the administration
LINE 295 maybe you mean heavy legs sensation?
LINE 343-346: there is much more literature now on this block.
Congratulations
Author Response
Dear reviewer,
Thank you very much for your comments.
We will proceed with the suggested modifications.
Best regards
The authors
Dear authors, thank you for the paper.
I read the paper with great pleasure. I have a few notes that I hope will provide insights to implement the quality of the work.
- Thank you so much for your comments. We are sure that suggested modification will implement the quality of our work.
INTRODUCTION
LINE 40: typing error (diffuse instead of diffused)
- Thanks, it has been changed in the text.
LINE 60: phrases could be linked better.
- Thanks, it has been modified in the text.
MATERIALS AND METHODS
2.1 PARAGRAPH: here are presented some results, for example the age and weight of the animals enrolled. I think that instead it should report some exclusion criteria.
- Thank you for the suggestion the manuscript has been modified. Age and weight have been included in the results and one exclusion criteria added in the M&M.
LINE 102: if the fluoroscopy used a dorso-ventral radiological beam why the position of the needle is confirmed by a latero-lateral fluoroscopic view?
- The text has been modified. Prior to injection it is important to carry out at least two views so the location of the needle in relation to other structures is confirmed. It is an additional method to confirm positioning but not the main one.
LINE 103: what caused the variations in the amount of contrast injected?
- Thanks for this interesting question. Any time the needle was moved an aliquot of contrast was injected and its pattern of diffusion analised, if the spreading was not considered correct the needle was moved and more contrast injected.
LINE 113: is not clear how the samples were chosen.
- The dyed tissue was resected and from this tissue 3 histological blocks were obtained.
LINE 133: typing error (staining instead of stained)
- Thanks, it has been changed in the text.
LINE 133-136: I would rephrase this section, as it is not very clear.
- Thanks, it has been changed in the text.
RESULTS
LINE 152: which direction followed the stain
- With the methodology used we cannot be completely sure about the direction of diffusion of the dye, but it looked like it diffused both cranially and caudally within the retroperitoneal space. If larger volumes of injectate are used, it will be easier to define the direction of the stain.
I would not put all the images as a subchapter of the results. Some of them are more related to the method part.
- We added the imaging in this section because it was the location allocated by the template found in the Animals journal submission form. We are very happy to modify the position of the pictures if it is possible.
DISCUSSION
LINE 196: there is some literature on the effects of prolonged opioids treatment also in veterinary medicine, maybe cite some of it.
- Thanks, a reference on effects of opioids in canine has been added.
LINE 198-200: I don’t see how this information are useful in this context
- We are happy to remove this sentence, but we consider important to underline the intensity of pain in dogs suffering of pancreatitis because the CPB or CPN can be used in subjects affected by this condition like it occurs in humans
LINE 261: Following the administration
- Thanks, this has been added in the text.
LINE 295 maybe you mean heavy legs sensation?
- Thanks, the text has been modified.
LINE 343-346: there is much more literature now on this block.
- We are aware, there are several studies looking at this technique, but we did not want to make the text too long adding information we did not consider essentials. The other reviewer suggested to reduce the words count.
Congratulations
- Thank you very much for your suggestions.
Round 2
Reviewer 1 Report
Comments and Suggestions for Authors
This is an interesting study with potential application for maller veterinary species of animals.
Both the Introduction and Discussion will require extensive editing .
The Introduction has far too much information related to the human species .We are dealing with pigs in this paper nd this must be the main topic of the discussion.
Line 80 --refernce to euthanasia --how were these animals killed?
Lines 144-147 describe the animals used. This correctly belongs in the methods section NOT the results
Line 200 The word "canine" canie is used incorrectly .It is an adjective and if used must be "canine species" or preferably as it is dogs use the word
Comments on the Quality of English LanguageMinor editing is required
Author Response
Dear reviewer,
thanks for your suggestions.
We are aware we refer especially to human literature when describing the CP anatomy (in the introduction, line 55-60) and the CPB and CPN efficacy and side effects in the discussion. There is no published literature available in swine or other veterinary species on this subject (apart the mentioned ones). In general use of fluoroscopy for interventional analgesia in veterinary subjects is scarce. We fully understand your concerns and this is why we add this in the limitation of the study. If you prefer us to remove some paragraphs to improve the manuscript quality please indicate us the one you think are more redundant and we will remove them.
line 80: the way subjects were euthanized has been included
line 144-147: it has been moved to the methods
line 200: the word has been change
thanks again to help us improve the quality of our manuscript
best wishes